# Association between national action and trends in antibiotic resistance: an analysis of 73 countries from 2000 to 2023

Peter Søgaard Jørgensen[1,2,3]*, Luong Nguyen Thanh[1,3], Ege Pehlivanoğlu[1], Franziska Klein[1¤a], Didier Wernli[4], Dusan Jasovsky[5¤b], Athena Aktipis[6], Robert R. Dunn[7], Yrjö Gröhn[8], Guillaume Lhermie[8¤c], H. Morgan Scott[9], Eili Y. Klein[10,11]

**1** Global Economic Dynamics and the Biosphere, The Royal Swedish Academy of Sciences, Stockholm, Sweden, **2** Stockholm Resilience Centre, Stockholm University, Stockholm, Sweden, **3** Uppsala Antibiotic Centre and Department of Women's and Children's Health, Uppsala University, Uppsala, Sweden, **4** Global Studies Institute, Transformative Governance Lab, University of Geneva, Geneva, Switzerland, **5** ReAct Europe, Uppsala University, Uppsala, Sweden, **6** Department of Psychology, Arizona State University, Arizona, United States of America, **7** Department of Applied Ecology, North Carolina State University, Raleigh, North Carolina, United States of America, **8** Department of Population Medicine and Diagnostic Sciences, Cornell University, Ithaca, New York, United States of America, **9** Department of Veterinary Pathobiology, Texas A&M University, College Station, Texas, United States of America, **10** One Health Trust, Washington, District of Columbia, United States of America, **11** Hopkins School of Medicine, Department of Emergency Medicine, Baltimore, Maryland, United States of America

¤a Current address: Social Sciences Group, Wageningen University & Research, Wageningen, The Netherlands
¤b Current address: Médicins Sans Frontières (MSF) International, Geneva, Switzerland
¤c Current address: Faculty of Veterinary Medicine, University of Calgary, Canada
* peter.sogaard.jorgensen@su.se

## Abstract

The world's governments have agreed on actions to address the challenge of antibiotic resistance. This raises the question of what level of national action is associated with improved outcomes, including both slower growth and lower levels of antibiotic resistance. Answering this question is challenged by variation in data availability and quality as well as disruptive events such as the COVID-19 pandemic. We investigate the association between level of national action and temporal trends in multiple indicators related to health system capacity, antibiotic use (ABU), absolute rates of resistance (ABR) and a Drug Resistance Index (DRI). Using the Global Database for Tracking Antimicrobial Resistance (TrACSS) to construct an index of national action, we apply cross-sectional regression across 73 countries to estimate the association between the level of action in 2016 and trends in national indicators (2000–2016). We find that national action is consistently associated with improved linear or categorical trends in all groups of indicators. Reductions are associated with a relatively high action index (range 0–4) for ABU (median 2.8, 25–75% quartile 2.6-3.3), ABR (3.0, 2.4-3.4), and DRI (3.5, 3.1-3.6). These associations are robust to the inclusion of other contextual factors related to socio-economic conditions, human population density, animal production and climate. Since 2016, a majority of both Low- and Middle-Income Countries (LMICs) and High-Income Countries (HICs) report increased action on repeated questions, while one third of countries report reduced action. The main limitations in interpretation are heterogeneity in data availability and in when actions have

**Data availability statement:** All code and data necessary to carry out the analyses are publicly available at https://github.com/PSJorgensen/national-actions-and-trends-in-antibiotic-resistance. Results can be viewed interactively at https://gedb.shinyapps.io/amr_trend.

**Funding:** We acknowledge funding from the Erling-Persson Family Foundation (P.S.J., L.N.T., E.P., F.K.), the European Union (ERC, INFLUX, 101039376, P.S.J.), the IKEA Foundation (P.S.J.), the Marianne and Marcus Wallenberg Foundation (P.S.J) and the Uppsala Antibiotic Centre (UAC, L.N.T.). The funders had no role in study design, data collection and analysis, decision to publish, or preparation of the manuscript. Views and opinions expressed are, however, those of the author(s) only and do not necessarily reflect those of the European Union or the European Research Council. Neither the European Union nor the granting authorities can be held responsible for them. We thank SESYNC for support for the Living with Resistance pursuit, which this paper is a product of.

**Competing interests:** The authors have declared that no competing interests exist.

been implemented. Our findings highlight the importance of national action to address the domestic situation related to antibiotic resistance and indicate the value of both incremental changes in reducing adversity of outcomes and the need for high levels of action in delivering reduced levels of resistance.

## Introduction

Antibiotic resistance (ABR) is a global public health challenge that in 2019 was estimated to contribute to 1.27 million deaths per year and was associated with a total of 4.95 million deaths [1]. In 2015, countries agreed a Global Action Plan to address the growing challenge of ABR, which was followed up in 2016 and 2024 with UN high-level meetings, where commitments were made to develop and implement national action plans (NAPs). The high burden of ABR and the increasing focus on national action highlights the importance of assessing the level of action that can be expected to lead to improvements in national conditions. However, such assessment is challenged by multiple factors, including the time it takes from policy adoption to effect of a policy, variation in data availability across countries, and the recent disruption to monitoring of the COVID-19 pandemic [2].

In 2016–17, the Global Database for Tracking Antimicrobial Resistance (TrACSS) completed its first survey of the presence and ambition of antibiotic resistance policies in more than 150 countries. Based on self-reporting, this database gives the first snapshot of the level of action taken to tackle antibiotic resistance around the world. Key indicators to consider the effects of policies on span from the upstream drivers that lead to increases in antibiotic use (ABU), ABU itself, levels of ABR and the exposure to and impact of resistant infections [3]. Among drivers, lack of health system capacity is a major contributor to inappropriate use [4,5]. For ABU, the total consumption of antibiotics adds selection pressures for resistance and the use of broad spectrum and last resort antibiotics are of particular concern [6]. For ABR, it is relevant to monitor effects of both common infections as well as resistance to last resort antibiotics. For exposure and impact, gaps in monitoring mean that good indicators are scarcely available [7].

In this paper we investigate the association of action reported in the TrACSS database with temporal trends in national indicators relating to antibiotic resistance from 73 countries (Fig 1). We apply a multi-indicator approach, covering indicators related to the health system, ABU, ABR and exposure to antibiotic resistant infections in humans (Drug Resistance Index, DRI).

## Methods

### Study design

We investigate national trends in indicators related to ABR by using a multi-indicator assessment approach according to a modified version of the Drivers-Pressure-State-Exposure-Effect-Action (DPSEEA) family of frameworks [8] (Fig 1, S1 Table, S2 Table, S3 Table). Such frameworks aim to assess how policies affects changes at multiple points in a system [3]. Similar frameworks, have been used to evaluate national progress on health and biodiversity issues [9], but also to understand the co-evolutionary dynamics of pesticide resistance [10].

Our main purpose is to understand to what degree self-reported policy action can help explain temporal trends in national indicators of drivers, pressures, state and exposure to antibiotic resistance, henceforth DPSE indicators (Fig 1A, see *Variables and Indicators* for details). Specifically, we construct an action index based on the first TrACSS survey in 2016–17.

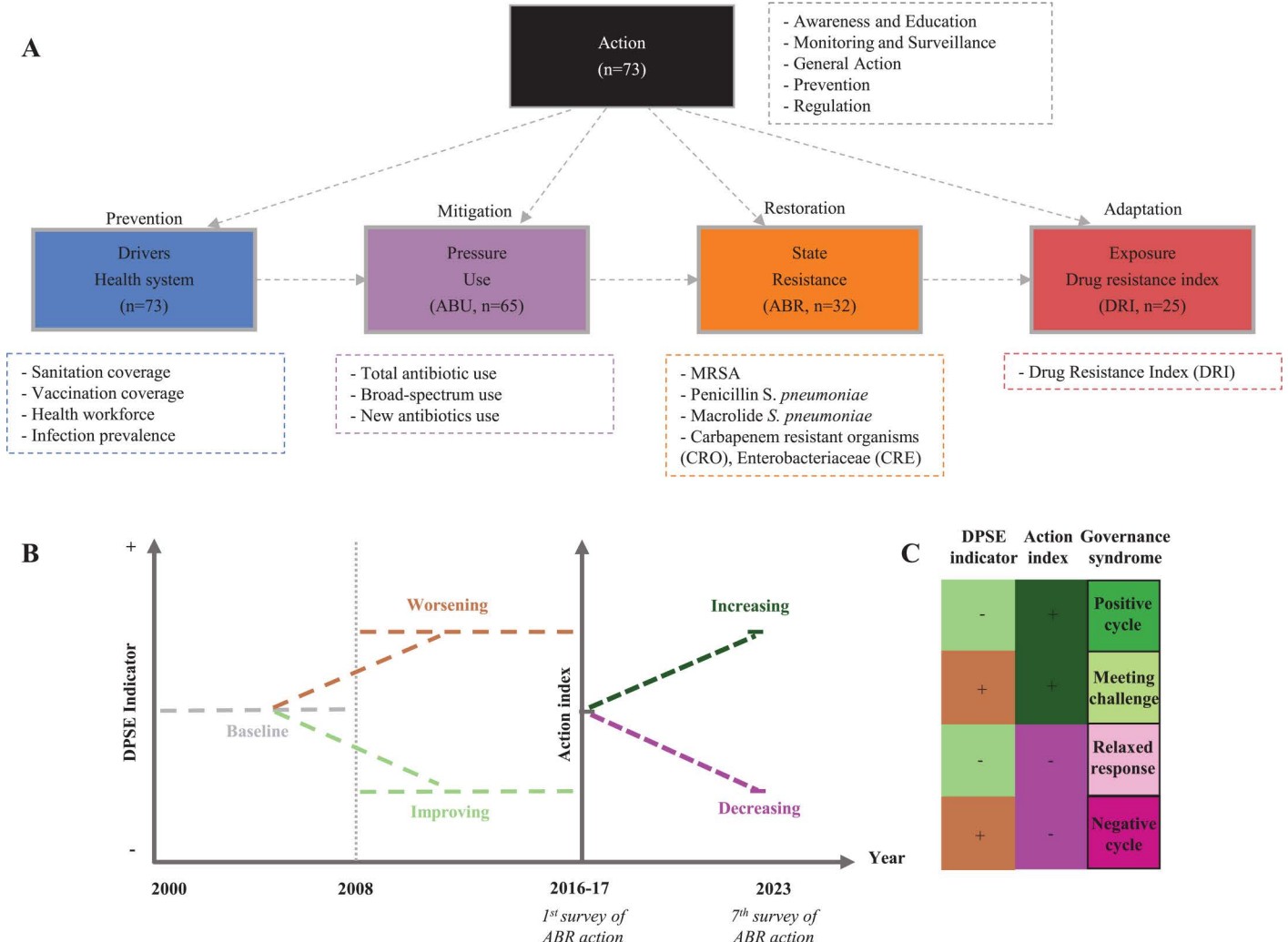

**Fig 1. Study design.** (A) DPSE indicators (tier 1) and their tier 2 components (boxes with dashed-lines). "n" refers to number of countries, for detailed information on indicators see S1 Table, S2 Table, S3 Table. (B) Temporal overview of analysis showing the two periods across which the trend in DPSE indicators is calculated and the timing of the first (2016-17) and seventh (2023) TrACSS survey. (C) Governance syndrome. Categorical trends in DPSE indicators and the action index combine to form four types of governance syndrome.

As 2016–17 was the first survey since the agreement of the Global Action Plan, the actions reported are assumed to mainly reflect measures implemented during the first decade and a half of the 21st century and to be relevant for temporal trends in indicators during this period. This is because countries will have had little time to take new large-scale action since 2015 and since any new action that has been taken since 2015, will have had little time to have a measurable effect. We therefore relate the action index to trends in DPSE indicators in the preceding 16 years (Fig 1B), specifically the change in indicators between the first and second half of the period 2000–2016. We also investigate the relative importance of action compared to covariates related to health systems, economy, human population, livestock production and climate (S4 Table). We apply cross-sectional regression and model selection to assess the strength of association with linear as well as categorical trends.

The above analysis is limited to the human health sector as monitoring in the animal sector is currently too sparse. To complement this analysis, we assess the degree to which production

of animal protein is correlated with level of policy action. We correlate the volume of production with level of action taken in the human and animal sector. Finally, we assess the extent to which changes in DPSE indicators are associated with subsequent changes in level of action from 2016 up until 2023 (Fig 1B). Here, a positive cycle is defined as a country experiencing decreases in DPSE indicators during the 2000–2016 period and subsequent increased action during the period 2016–2022 and a negative cycle if the DPSE indicator increased and action decreased. Increases in DPSE indicators and increased action were categorized as "Meeting the challenge" and decreases in DPSE indicators and decreased action were categorized as "Relaxed responses" (Fig 1B).

## Setting

A total of 73 countries were included in the analyses (S5 Table). The countries represent all inhabited continents, in order of sample size: Europe (n = 35), Asia (n = 20), South America (n = 7), Africa (n = 5), North America (n = 4), Oceania (n = 2). These were divided between 37 High-Income Countries (HICs) and 36 Low- and Middle-Income Countries (LMICs).

## Variables and indicators

DPSE categories capture indicators of health system development (Drivers), ABU (Pressures), ABR (State), and antibiotic resistance relative to use of an antibiotic as measured by the DRI (Exposure). Composite DPSE indicators vary in the number of levels (tiers) at which they can be disaggregated, from one (Exposure), over two (Pressure and State), to three (Drivers), for additional details see S1 Text.

**Drivers – health system.** We included fifteen variables relating to the general state of the human health system across four tier 2 indicator groups: infection prevalence (Infection), sanitation standards (Sanitation), vaccination coverage (Vaccination), and health care workforce (Workforce). Each consists of two to four tier-3 indicators (S1 Table).

**Pressure – antibiotic use (ABU).** We include antibiotic consumption data for humans for the years 2000–2015. Data are obtained for three tier-2 indicators: use of six broad-spectrum antibiotics [11], measured as percent of total use; use of 25 newly available antibiotics [12]; and total per capita use (total), measured as the Daily Defined Dose (DDD) per 1000 inhabitants per day (S2 Table).

**State – absolute rates of resistance (ABR).** We include three groups of tier-2 indicators for antibiotic resistance: Methicillin-resistant *Staphylococcus aureus*, Carbapenem resistance in Enterobacteriaceae (CRE) and in other bacteria (CRO), and Streptococcal resistance to macrolides and penicillin (S2 Table).

**Exposure – relative rates of resistance (DRI).** We use the DRI as an indicator of exposure as it takes into account resistance relative to the use of antibiotics in a country, both of which are important factors in determining likely exposures to resistant infections [13]. For this index we include 16 drug and bacterial combinations including the bacteria: *Enterococcus faecalis/faecium*, *Escherichia coli*, *Klebsiella pneumonia*, *Pseudomonas aeruginosa* and *Staphylococcus aureus* (S3 Table). Although the DRI index has been critiqued previously as a standalone indicator [14], we here use it as a part of a multi-indicator analysis to complement patterns in absolute rates of use and resistance.

**Action – self-assessment survey.** Selected questions from the AMR TrACSS dataset (S6 Table) were divided into five thematic categories: Awareness & Education, Monitoring & Surveillance, Prevention, Regulation, and General questions(S7 Table). For each country, action scores within categories were calculated by a simple average with individual responses ranging from 0 to 4, in order of increasing ambition. The action index was calculated as the

average of all five categories equally weighted. The action index used for the governance syndrome analysis include questions that were asked repeatedly over the years 2016–17 to 2022–23 (S6 Table). Here we calculated the difference between the last year of response (2022–23 for all except 2021–22 for Romania) and the first year of response (2016–17).

## Data sources

A variety of data sources was used for measuring the DPSE indicators (S1 Table, S2 Table, S3 Table), explanatory covariates (S4 Table), and the action index (S6 Table). Data for health system drivers came from the United Nations, the World Bank, and the World Health Organization. Data for ABU were obtained the IQVIA database [15] for the years 2000–2015. Data for ABR came from ResistanceMap [16]. Data sources for the DRI were identical to those for ABU (IQVIA [15]) and ABR (ResistanceMap [16]). Data for the action index came from the Global Database for TrACSS survey in 2016–17 and 2022–23. Data for covariates came from The Eora Global Supply Chain Database (GDP), World Income Inequality Database – WIID (Gini index), NASA Center for Climate Simulation – BioClim (mean temperature), Gridded Livestock of the World – GLW (animal production), and NASA's Socioeconomic Data and Applications Center – SEDAC (human population density).

## Bias

Countries are only included if they have sufficient data for the Drivers indicator group and at least one of the other DPSE indicators (ABU, ABR or DRI). This reduces the number of LMICs that can be investigated and biases the sample towards well monitored, often HICs.

## Study size

Countries included in the analyses were filtered based on two criteria. First, answering the TrACSS survey in 2016–17 (S6 Table). Second, reporting at minimum three years in the period 2000–2008 as well as in 2008–2016 for ABU or ABR indicators. Applying these criteria, we are able to analyse 73 countries in total (drivers n = 73, ABU n = 65, ABR n = 35, DRI n = 25, Fig 1A).

## Quantitative variables

**Linear and categorical trends.** We calculated the average for each DPSE indicator at tier 1–3 (S1 Table, S2 Table, S3 Table) across the years 2000–2008 (henceforth baseline) and 2008–2016. First, raw data were standardized by their standard deviation (s.d.= 1), which allow for comparing indicators that use different units of measurement. Then the national changes in means between two periods was calculated as:

$$(1) \quad Linear\ trend = \overline{X}_{2008-2016} - \overline{X}_{2000-2008}$$

Henceforth referred to as the *linear trend* (abbreviated l.t.). We also analyzed countries based on the sign of change of the linear trend (increasing vs. decreasing), henceforth referred to as the *categorical trend* (abbreviated c.t.).

## Statistical method

**Model formulation.** We fitted gaussian mixed effect regressions models with the linear trend (equation 2) and the action index as response variable (equation 3), the latter to assess the ability of the categorical trend to explain variation in action (S8 Table). We also fitted binomial mixed effect models with the proportion of declining tier 2 indicators as the

response variable, weighted by the availability of indicators (equation 4, S9 Table). All models included baselines (*Baseline*) as covariates and equation 2 and 3 a random grouping variable (*Income*) indicating income level (HIC vs LMIC).

(2) *Linear trend ~ Action + Baseline + (1|Income)*

(3) *Action ~ Categorical trend + Baseline + (1|Income)*

(4) *Proportion in decline ~ Action + Baseline*

To explore how action various with size of animal production we also fit a model with action as response variable and action in the human and animal health sector as explanatory variables. For this analysis, we used questions that were asked in parallel for the two sectors (S6 Table).

**Sensitivity analysis.** We applied model selection to assess the relative importance of action variables compared to other covariates (S10 Table). Here we fitted gaussian models to the linear trend and binomial models to the categorical trend and we included countries as random effects, the type of DPSE indicator and country income level as factorial variables and covariates relating to the health system, economic condition, human population, animal production mass, and annual mean temperature (S11 Table). We also allow for pairwise interactions between factorial variables and covariates. All global model formulas are detailed in S11 Table. We tested the sensitivity to various combinations of country and indicator subsets, resulting in a total of 17 model selection procedures for the linear trend and 16 for the categorial trend where we did not fit models to the Exposure data as there was too little variation in the response variable outcome. We used the R package "MuMIn" [17] and applied the Akaike's information criterion corrected for small sample sizes (AICc). We report averaged effects for the 95% Akaike weighted subset as well as coefficients from the models with lowest AICc. All statistical analyses were carried out in R 4.3.1 [18].

## Results

### Trends in DPSE indicators

The four tier-1 DPSE indicators exhibit mixed 16-year trends with health system drivers improving (linear trend = -0.126 ± 0.017 s.d., p < 0.001, categorical trend = 6/73 countries increasing, p < 0.001), ABU and DRI increasing (ABU: l.t. = 0.289 ± 0.048, p < 0.001, c.t. = 55/65, p < 0.001; DRI: l.t. = 0.184 ± 0.065, p = 0.01, c.t. = 21/25, p = 0.002), and non-significant trends in ABR (l.t. = 0.019 ± 0.069, p = 0.8, c.t. = 16/32, p = 1). The increase in ABU is also seen for three of the four tier 2 indicators (total per capita use: l.t. = 0.329 ± 0.065 p < 0.001, c.t. = 50/65, p < 0.001; broad-spectrum antibiotics: l.t. = 0.278 ± 0.069, p < 0.001, 47/65, p = 0.001; newly available antibiotics: l.t. 0.232 ± 0.061, p < 0.001, c.t. = 55/63, p < 0.001). The only tier-2 ABR indicator that showed a consistent increase over time across countries was resistance to last resort carbapenems (c.t. = 20/28, p = 0.028).

### Association between DPSE trends and action

At the tier-1 level, action is negatively correlated with three of four DPSE indicators for both linear and categorical trend (Fig 2, S12 Table, S13 Table). For the linear trend, action is negatively associated with health system drivers, ABU and DRI (Fig 2A, Fig 2B, Fig 2D, p < 0.05). For the categorical trend, action is negatively associated with ABU, ABR and DRI (Fig 2F, Fig 2G, Fig 2H, p < 0.05).

Complementary analyses indicate that the overall explanatory ability of the action index is (a) not due to a correlation with the baseline state of the indicators (S14 Table), is (b) highest

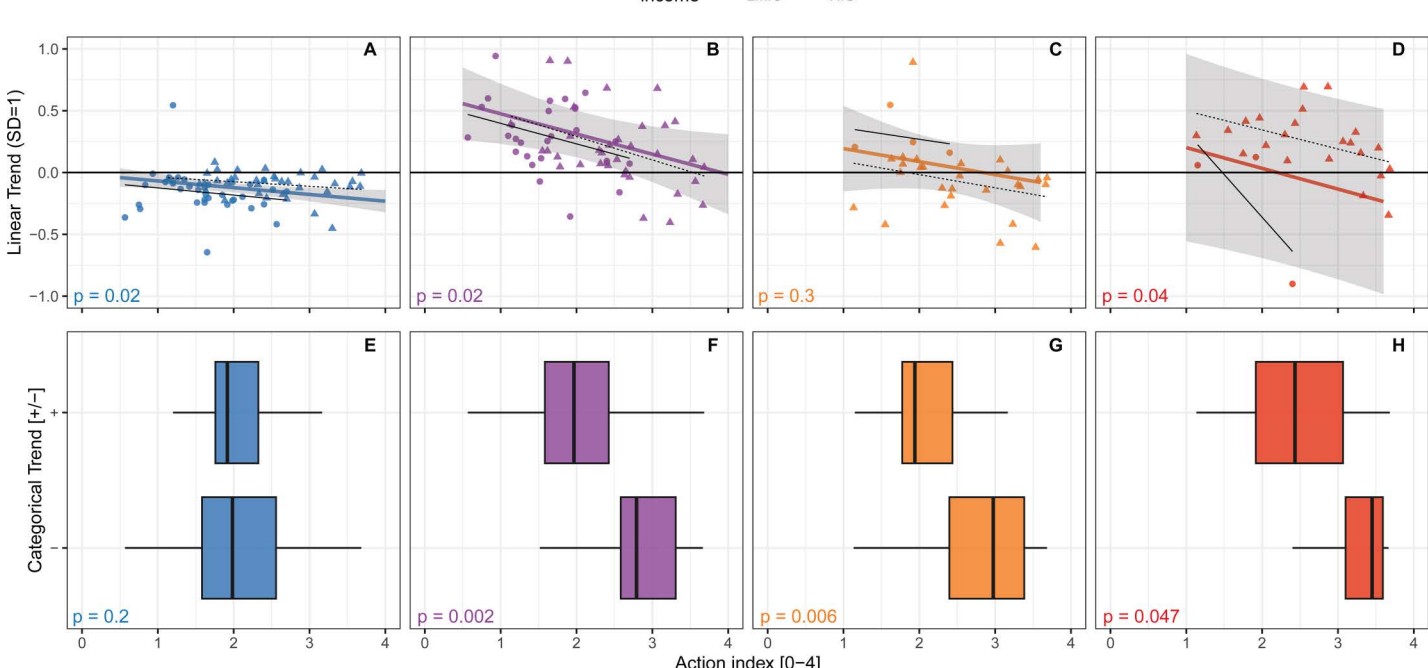

**Fig 2. Association between action index and linear trend (A, B, C, D), and categorical trend (E, F, G, H).** Indicator p-values are from linear mixed models with country income group as random effect. For detailed formulas, see S8 Table. Blue represents drivers of antibiotics resistance, purple represents antibiotics use, orange represents resistance, and red represents DRI.

at the level of tier 1 (S12 Table, S13 Table), and (c) outperforms individual components of the action index (S15 Table, S16 Table, S17 Table, S18 Table, S19 Table, S20 Table, S21 Table, S22 Table, S23 Table, S24 Table). Thus, individual action components are on average associated with 1 (linear trend) and 1.4 (categorical trend) DPSE indicators, respectively. Of these, the Monitoring and Surveillance component is most commonly associated with DPSE indicators (two of four indicators in both types of models) while Prevention is not associated with any indicators.

The level of action required for consistent negative categorical trend varies widely between DPSE categories (Fig 3, S25 Table, S26 Table). Here, DRI and ABU exhibits the largest needs for action, converging around an action index of 3.5-3.6 out of 4 for 50% probability (Fig 3). DRI showed signs of a threshold behaviour, with very low probabilities of improvement with action below 2 and 50% chance of improvement with an action index above 3.5. For ABU, reduction in 25% of variables was achieved at an action score of 2.5 and 50% at 3.5. We found a 30% chance for reduction in ABR at 1.5 and 50% chance at 2.5. Drivers show overall high probabilities of improvement irrespective of action, but with a slight negative trend as action increases.

## Model selection

Action variables had relatively high levels of importance in model selection usually ranking in top 3 (Fig 4, S1 Fig, S27 Table), and with consistent negative associations with worsened outcomes (S2 Fig, S3 Fig). For linear trends, the general action variable often had a higher importance score than the action index. Here, health system indicators such as workforce were also important and associated with increasing trends for ABU, ABR and DRI. For categorical trends, the action index repeatedly featured in the best selected models (negative coefficients) along with animal

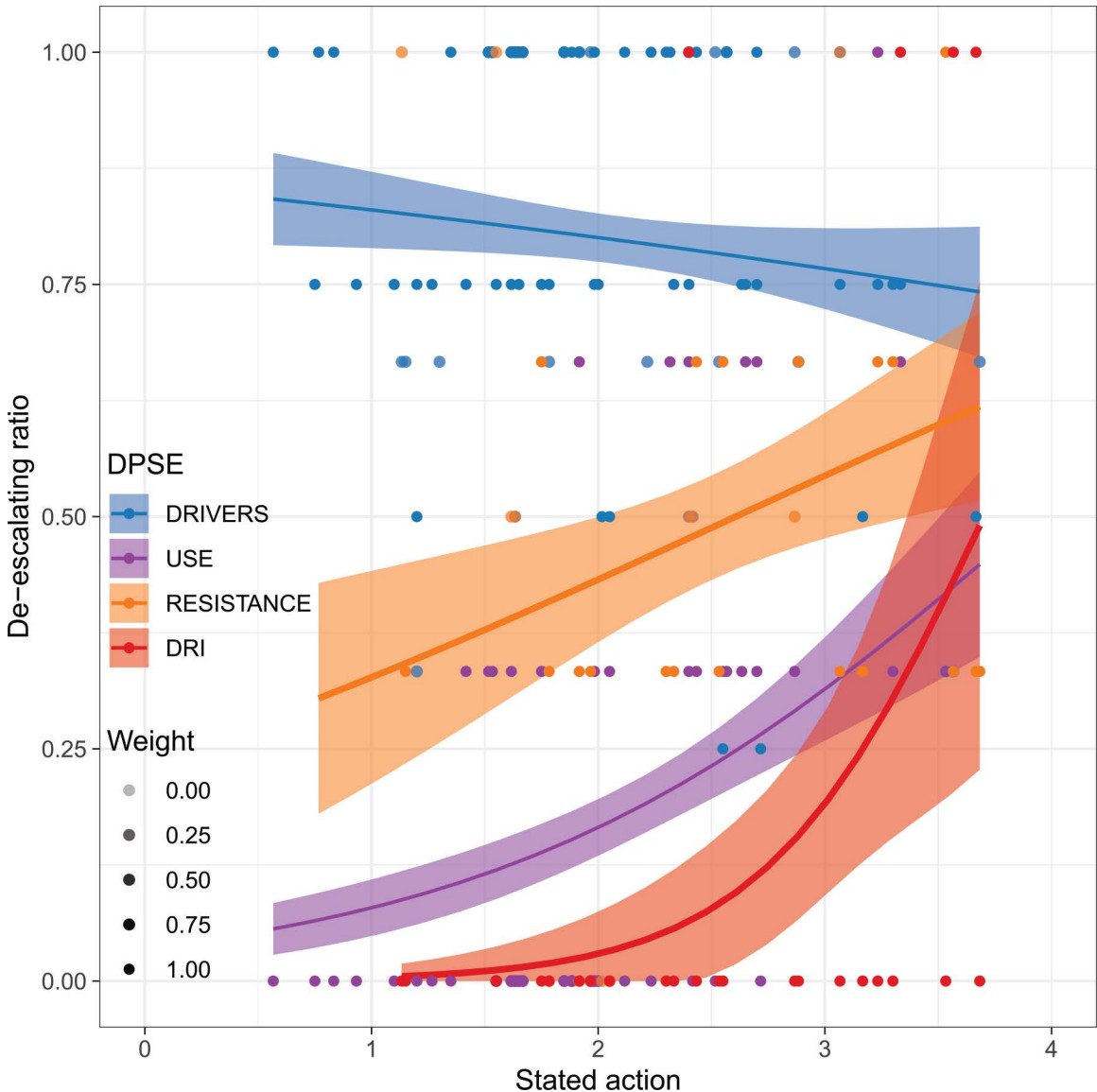

**Fig 3. Proportion of declining indicators as a function of the action index.** Shown is the proportion of lower-level indicators within a category that have witnessed a reduction from 2000 to 2016. Weight refers to the proportion of variables avaiable for a given country. Uncertainty bands indicate standard errors.

production (positive coefficients) and mean temperature (interaction with DPSE) (S3 Fig). Similar effects were also visible when limiting the analysis to HICs, but almost absent within LMICs, potentially due to their larger heterogeneity and data scarcity (S4 Fig, S5 Fig, S6 Fig).

## Action in the animal vs human health sector

Countries with larger animal production generally take more ambitious action on animal as well as human health specific issues (slope = 0.60, s.e. = 0.1, t = 6.1, p < 0.001, Fig 5). This is true especially in HICs, where animal health specific action was more sensitive to tonnage of animal protein produced than in LMICs (slope difference = -0.61, F = 12.6, df = 129, p < 0.001, Fig 5). Thus, in HICs, large producers of animal

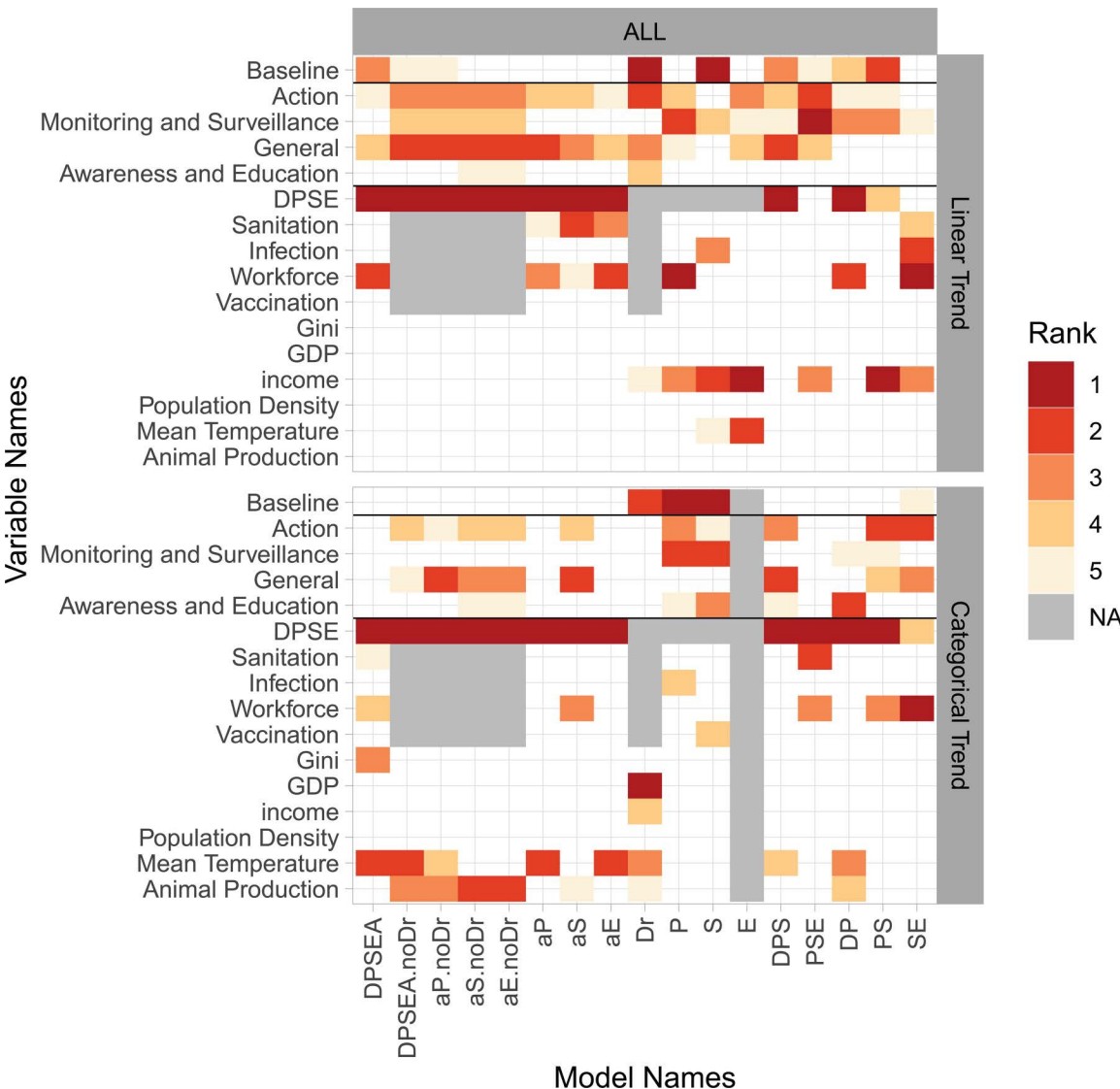

**Fig 4. Most important variables in model selection.** The rank of the five most important variables (rows) is shown using color coding. Each column represents a unique model selection procedure on the linear trend (17 procedures) or the categorical trend (16 procedures). Model names refers to the subset of DPSE indicators included D (Driver), P (Pressure), S (State), E (Exposure). In model names, "noDr" indicates exclusion of health system variables as explanatory variables and "a*X*" refers to analysis of DPSE for country subsets with *X* variable available. See S10 Table, S11 Table for details on each model selection procedure.

protein had a higher action index on specific animal health questions than they had for human health questions, whereas small producers had a higher action index for the human health questions (slope difference = -0.52, p = 0.01, df = 80, F = 6.2, Fig 5A). In LMICs on the other hand, the human action index was generally higher than the animal action index (mean difference = 0.25, p = 0.02, df = 179, F = 5.8, Fig 5B) with both increasing for bigger producers (slope = 0.29, p < 0.001, df = 179, t = 5, Fig 5B). The level of action in the largest LMIC producers is significantly lower than in the largest HIC producers pointing to a need for further strengthening action in several of the world's largest producing countries (Fig 5).

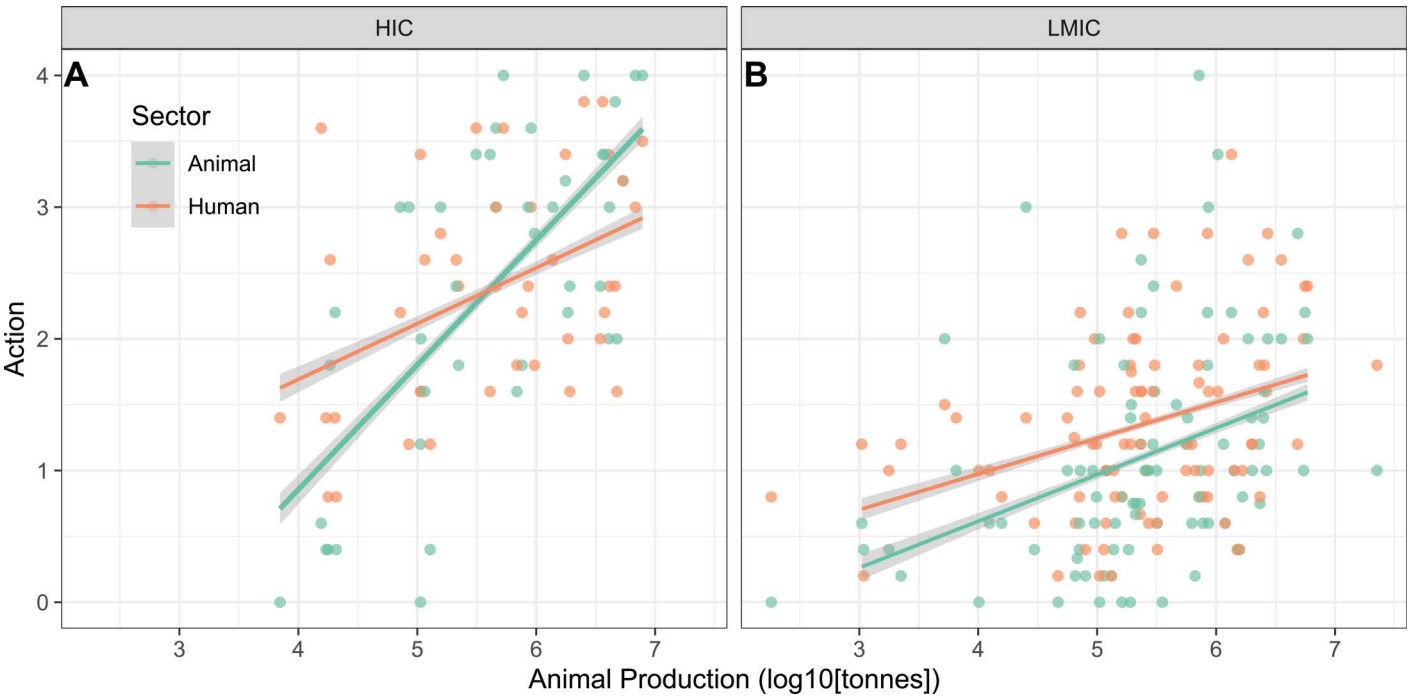

**Fig 5. Animal production and the action index.** Action level of countries on human (red) and animal (blue) specific issues in relation to total production of vertebrate biomass (mammals, birds and fish) for high-income (HIC) and low- and middle-income countries (LMIC).

### Changes in action over time

Three quarters (76%) of countries analysed for association with DPSE indicators in 2016–17 report increased action between 2016–17 and 2023. This is the case in both HICs (70%) as well as in LMICs (83%). However, almost 25% of countries lowered their ambition including almost 30% of HICs (all = 23%, HIC = 30%, LMIC = 17%, Fig 6). For drivers, we see that more than half of HICs and LMICs are in a positive cycle with improved conditions and subsequent increased policy ambition. However, this proportion drops markedly for ABU, ABR, and DRI. For ABU in LMICs, 3 out of 31 countries (10%) are in a positive cycle and 6 in a negative cycle, but 70% are meeting the challenge of increasing ABU with increased policy responses. For HICs, negative cycles are fairly frequent for ABU (24%, n = 34) and DRI (32%, n = 22) and positive cycles very rare for DRI (5%, n = 22).

### Discussion

Our results show associations between self-reported action on antibiotic resistance in 2016–17 and trends in the preceding 16 years in health system drivers, ABU, ABR and DRI. These results are robust to inclusion of other contextual variables in multivariate models, strongest at the aggregate scale of both DPSE and action indicators, and not due to a correlation with the baseline state of indicators. The results indicate that every step of improved action is important to reduce the magnitude of increases in ABU and DRI (Fig 2B, Fig 2D), and that high levels of action can help achieve reductions in ABU, ABR and DRI (Fig 2F, Fig 2G, Fig 2H). Here we discuss the interpretation of the results, some potential limitations as well as consequences for policy.

### National action and DPSE indicators

National action on antibiotic resistance has been critiqued for not being well enough funded and national action plans (NAPs) for not being fully developed and operationalized, with

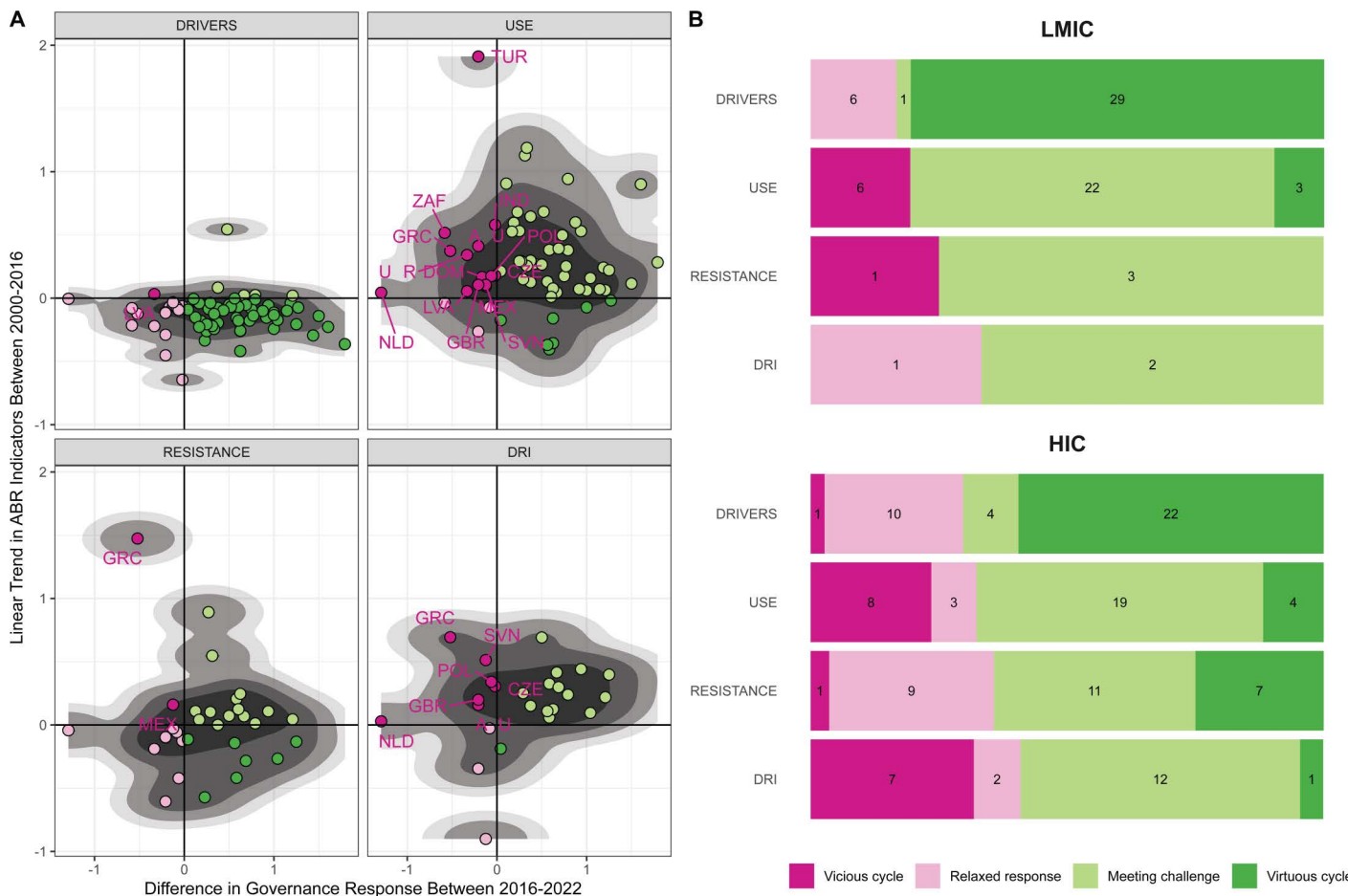

**Fig 6. Governance syndrome.** (A) Trend in DPSE indicators as a function of change in action index. Countries in negative cycle are named with ISO3 codes (see S5 Table). (B) Distribution of governance syndrome by DPSE indicator and country income level.

some indications of adoption of standard templates [19–22]. These concerns about the state of national action increase the uncertainty and potential discrepancy between data reported and action taken on the ground. We would expect, such default, template responses to make it harder to detect an association between reported actions and DPSE indicator trends. That is because, such responses make it harder to distinguish between countries with well-implemented and poorly-implemented policies. As we compare trends in DPSE indicators leading up to the 2016–17 survey, some recently taken action will likely not have had an effect on the trends. On the other hand, some policies will have been implemented for several years prior. That we detect these associations even with the above-mentioned uncertainties indicates that well implemented action probably has fairly large and measurable effects with the time needed for achieving the intended effect being the major unknown. It also implies that the estimated levels of action needed to reduce ABU, ABR, and DRI, given sufficient time, are likely safe levels and that reductions might be achievable at lower levels.

We find that the action index more consistently explains univariate changes in the four composite DPSE indicators than any of the individual components of the index and that the explanatory ability of the action index is most consistent at the highest level of aggregation of the DPSE indicators. These results indicate the importance of assessing aggregate policy

responses and evaluating their potential impacts on combined indices of multiple indicators. This approach helps account for cross-country variation in social, economic and environmental context and related variation in needs for action. Monitoring and surveillance and general action (such as early progress on NAP development) also explain variation in trends in several multivariate models. Here, general action likely indicates the importance of responding systemically through coordinated responses across sectors e.g., through NAPs [21,23]. The monitoring and surveillance variable performs well in model selection for several models limited to ABU and ABR well as models limited to HICs. The importance of this variable could indicate its importance in enabling informed decision making as well as the adaptive management of both the use of antibiotics and containment of resistant variants [24].

DRI is the hardest of the indicators to achieve reductions in, followed by ABU and ABR. Taken together, we interpret these patterns as indications that absolute rates of resistance can more easily be mitigated by e.g., cycling antibiotics, while reducing use requires a wider infection prevention and control strategy. Reducing relative rates of resistance may also require improving use of antibiotics across the board to mitigate selection pressures on resistance e.g., when a new antibiotic is adopted. The linear trend in ABR shows fairly weak correlation with action, whereas the categorical trend is consistently correlated. The former pattern could be an indication of (1) a non-linear association between action and ABR; (2) that bacteria targeted for management vary by country and potentially go beyond the three indicators available for this analysis, or; (3) that absolute rates of resistance can be circumvented through various aforementioned strategies. The latter pattern could indicate a threshold of action above which such reductions are more likely to be achieved. Overall, national action has a fairly small effect size on the linear trend in health system drivers at ca. 0.25 s.d. across the range of the action index compared to 0.5 s.d. for ABU and DRI. This could be because health system improvements are driven by other policy instruments rather than AMR specific policy. The weakly negative association between action and proportion of improving drivers might be due to such improvements being much harder to achieve for countries with more developed health systems, i.e., countries where almost the entirely population has access to sanitation and vaccines, infection rates are low and where health care workforces are already high.

## Health systems, climate and animal production

Model selection reveals that multiple contextual factors help explain additional variation in DPSE indicator trends. These include variables related to health system development, climate and animal production. The relationship between health systems and antibiotic resistance is multifaceted. While the highest burden of antibiotic resistance is often due to lack of access to healthcare and antibiotics, health systems are also associated with higher levels of regulated use [25–27]. These consequences for equitable access might be driving the positive association between health workforce and the linear trend in ABU, ABR and DRI in multivariate models.

A positive association between warmer mean temperatures and the categorical trend of DPSE indicators is likely not due to collinearity with GDP per capita, inequality, or country income level as these are all considered as covariates (Fig 4). Possible explanations include a potential non-linear effect of climate on biological factors such as bacterial growth rates [28] or horizontal gene transfer [29]. Alternatively, one or more social, economic or environmental factors that we were not able to include in this analysis could be correlated with our climate indicator.

Animal production is positively associated with the categorical trend of DPSE indicators in four models that do not include health system indicators as explanatory variables (Fig 4). While the importance of animal production is lower when health system variables are

included, the pattern is still worth examining given the potential concern about any spill-over effect from the animal sector to the human health sector. Multiple explanations can be hypothesized for such patterns, including spillover of residues, resistance genes, or cultural factors relating to antibiotic use, to name a few [30]. Whatever any potential causal explanations might be, these patterns indicate that additional action in both the animal and human sectors may be needed to achieve indicator reductions in large animal producing countries. This challenge should not be underestimated as countries with large animal production are already reporting higher levels of actions in both the human and animal health sector (Fig 5). Reducing the variation in action across countries by elevating the lower level of action of large producers could help address any spillover effects.

## Changes in action

While most countries increase their level of action in 2023 compared with 2016–17, around a quarter reduce their levels of action. Reductions in the action index might reflect reduced prioritisation of the policies in the ABR area e.g., during and after the pandemic [2], but might also be due to enhanced accuracy of the data reported in the TrACSS survey compared to the first years. Improved accuracy in reporting would influence findings regarding governance syndromes by inflating the number of countries in the 'relaxed response' or 'negative cycle' categories. Given the high levels of action associated with reductions of ABU and DRI, it is expected that positive cycles are the least common in these two categories. That is in part due to countries with high levels of action that reduced ABU and DRI having less options to improve their score in 2023. Going forward it will be important to find governance mechanisms, that enable countries to progress in taking new action while focusing on implementing and securing sustainable funding for currently planned actions.

## Limitations

Our study has several limitations that should be taken into account when interpreting the results. First, our study is limited by the data availability in countries, which vary by indicator group and is generally biased towards HICs. Care should be taken when interpreting the results to not transfer them outside of their context. For application to specific income-settings, results relating to the specific HIC model and LMIC model should be used. The level of action is self-reported by countries and does not necessarily reflect the actual level of policies implemented in the country. Further, we cannot account for how long policies have been implemented. For example, it is reasonable to assume that countries with higher policy ambition levels have been frontrunners in the field for some time, policies therefore have been in place longer, and potentially showing a larger effect compared to countries who have recently implemented such policies. Our data is limited to antibiotic indicators in the human sector and does not necessarily say anything about the level of action needed for reducing adversity or improving trends in crop or animal health systems.

## Conclusions

In conclusion, our analysis indicates the importance of governments taking ambitious and comprehensive action on antibiotic resistance across sectors to improve the national situation. At the same time, in cases where reductions cannot yet be achieved, each additionally implemented action is likely to help reduce the severity of trends in antibiotic use and relative rates of resistance, stressing the need for continuously increased action. Future studies should assess the association between national action and trends in the DPSE indicators during the time period following the agreement of the Global Action Plan.

## Supporting information

**S1 Text. Supplementary Methods.**
(PDF)

**S1 Fig. Importance Scores of Variables in Model Selection from Averaged Models.**
(PDF)

**S2 Fig. Coefficient Estimates of Averaged Models for All Countries in Linear Trend and Categorical Trend Models.**
(PDF)

**S3 Fig. Coefficient Estimates of Variables from Best Selected Models from Model Selection.**
(PDF)

**S4 Fig. Rank of Variables in Averaged Models for Different Income.**
(PDF)

**S5 Fig. Importance Scores of Variables in Model Selection from Averaged Models for Different income Groups.**
(PDF)

**S6 Fig. Coefficient Estimates of Averaged Models for Countries with Different Income Groups.**
(PDF)

**S1 Table. Indicator selection for Driver categories.**
(PDF)

**S2 Table. Indicator selection for Use and Resistance categories.**
(PDF)

**S3 Table. Indicator selection for DRI (exposure) category.**
(PDF)

**S4 Table. Ecological variables used as covariates.**
(PDF)

**S5 Table. List of countries included in the study.**
(PDF)

**S6 Table. Governance Syndrome questions.**
(PDF)

**S7 Table. Questions used for calculating the action index.**
(PDF)

**S8 Table. Model Formulas for Association between Action and Indicator Linear Trend and Categorical Trend.**
(PDF)

**S9 Table. De-escalation plot formulas for univariate models.**
(PDF)

**S10 Table. Global Models Data Subset Formulas for The Model Selection.**
(PDF)

**S11 Table. Model selection global model formulas.**
(PDF)

**S12 Table. Association Between Linear Trend and Action.**
(PDF)

**S13 Table. Association Between Categorical Trend and Action.**
(PDF)

**S14 Table. Association Between Baseline and Action.**
(PDF)

**S15 Table. Linear Trend and Awareness and Education.**
(PDF)

**S16 Table. Categorical Trend and Awareness and Education.**
(PDF)

**S17 Table. Linear Trend and General.**
(PDF)

**S18 Table. Categorical Trend and General.**
(PDF)

**S19 Table. Linear Trend and Monitoring and Surveillance.**
(PDF)

**S20 Table. Categorical Trend and Monitoring and Surveillance.**
(PDF)

**S21 Table. Linear Trend and Prevention.**
(PDF)

**S22 Table. Categorical Trend and Prevention.**
(PDF)

**S23 Table. Linear Trend and Regulation.**
(PDF)

**S24 Table. Categorical Trend and Regulation.**
(PDF)

**S25 Table. De-escalation Merged Model Comparison Results.**
(PDF)

**S26 Table. De-escalation Merged Model Results.**
(PDF)

**S27 Table. Model selection table including variables for best selected models and null models.**
(PDF)

## Acknowledgments

The manuscript is a product of the SESYNC Pursuit, Living with Resistance, we thank all participants for their contributions to discussions during the meetings.

## Author contributions

**Conceptualization:** Peter Søgaard Jørgensen, Didier Wernli, Dusan Jasovsky, Athena Aktipis, Robert R. Dunn, Yrjo Gröhn, Guillaume Lhermie, H. Morgan Scott, Eili Y. Klein.

**Data curation:** Luong Nguyen Thanh, Ege Pehlivanoglu, Franziska Klein, Eili Y. Klein.

**Formal analysis:** Peter Søgaard Jørgensen, Luong Nguyen Thanh, Ege Pehlivanoglu, Franziska Klein, Eili Y. Klein.

**Funding acquisition:** Peter Søgaard Jørgensen.

**Investigation:** Peter Søgaard Jørgensen, Ege Pehlivanoglu, Franziska Klein, Didier Wernli, Dusan Jasovsky, Athena Aktipis, Robert R. Dunn, Yrjo Gröhn, Guillaume Lhermie, H. Morgan Scott, Eili Y. Klein.

**Methodology:** Peter Søgaard Jørgensen, Didier Wernli, Robert R. Dunn, Yrjo Gröhn, Guillaume Lhermie, H. Morgan Scott, Eili Y. Klein.

**Project administration:** Peter Søgaard Jørgensen, Ege Pehlivanoglu.

**Resources:** Eili Y. Klein.

**Supervision:** Peter Søgaard Jørgensen.

**Visualization:** Peter Søgaard Jørgensen, Luong Nguyen Thanh, Ege Pehlivanoglu, Franziska Klein.

**Writing – original draft:** Peter Søgaard Jørgensen, Luong Nguyen Thanh, Ege Pehlivanoglu.

**Writing – review & editing:** Peter Søgaard Jørgensen, Luong Nguyen Thanh, Ege Pehlivanoglu, Didier Wernli, Dusan Jasovsky, Athena Aktipis, Robert R. Dunn, Yrjo Gröhn, Guillaume Lhermie, H. Morgan Scott, Eili Y. Klein.

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
