## [Decision Letter · Decision Letter 0]

3 Sep 2024

PGPH-D-24-01686

Association between national policy and trends in antibiotic resistance: an analysis of 73 countries from 2000 to 2023

Dear Dr. Søgaard Jørgensen,

Thank you for submitting your manuscript to PLOS Global Public Health. After careful consideration, we feel that it has merit but does not fully meet PLOS Global Public Health’s publication criteria as it currently stands. Therefore, we invite you to submit a revised version of the manuscript that addresses the points raised during the review process.

We look forward to receiving your revised manuscript.

Kind regards,

Ashish KC

Academic Editor

Journal Requirements:

1. We ask that a manuscript source file is provided at Revision. Please upload your manuscript file as a .doc, .docx, .rtf or .tex.

Additional Editor Comments (if provided):

Reviewers' comments:

Reviewer's Responses to Questions

**Comments to the Author**

1. Does this manuscript meet PLOS Global Public Health’s publication criteria ? Is the manuscript technically sound, and do the data support the conclusions? The manuscript must describe methodologically and ethically rigorous research with conclusions that are appropriately drawn based on the data presented.

Reviewer #1: Yes

Reviewer #2: Yes

2. Has the statistical analysis been performed appropriately and rigorously?

Reviewer #1: Yes

Reviewer #2: Yes

3. Have the authors made all data underlying the findings in their manuscript fully available (please refer to the Data Availability Statement at the start of the manuscript PDF file)?

Reviewer #1: Yes

Reviewer #2: No

4. Is the manuscript presented in an intelligible fashion and written in standard English?

Reviewer #1: No

Reviewer #2: Yes

5. Review Comments to the Author

Reviewer #1: Thank you very much for inviting me to review this interesting work on assessing the national government reporting on AMR action and the change in terms of prevention, mitigation, restoration and adaptation. I must applaud the self-explanatory figures used to explain the process and analysis. However, I have major concern on the clarity of paper been written, presented and the conclusion made.

Abstract

I could not understand what DPSEEA framework is as it was provided for the first time. The study design is not avialable as well as the number of countries included in the analysis. How were the countries selected, I guess it was based on the TrACCS. What is the source of government reported policy action and the source of different indicators.

I could not understand the research question until I saw figure 1. Can authors keep it simple

Association of government reported AMR action on the prevention, mitigation, restoration and adaptation. OR Association of government reported AMR action on the health system strengthening, anti-microbial use, anti-biotic resistance and drug resistance.

In the abstract, the second and third sentence is confusing and does not provide good information of the work you are doing, can you please remove it.

Main text

Introduction- The first paragraph is well written and explains the context. I got lost while reading the second paragraph. In the second para you mention drivers and indicator. Can you please provide the name of indicator. I think there should be a good amount of explanation of four indicators.

The third paragraph is more of method, it should be in method section. Also, please provide the full form such as DPSEEA, as the acronym is used for the first time.

Please provide your research objective clearly, the research objective as in abstract is not well stated.

Method

Can you please use a STROBE checklist to ensure that you do not miss the manuscript reporting. For example, the design is missing. Study setting- The different countries included are missing

Selection criteria

Framework- Figure 1 can be one.

Data analysis

I am confused why you did an association of action done in 2016 as exposure while the health system strengthen, ABU, AMR as outcome. The reason I can asking this, the Global action plan was endorsed in 2015 and reporting of action was done a year after. How can a any action taken in a year, change the vaccination coverage, sanitation coverage, ABU as these are systematic level changes that requires an optimal years duration. Can you do the validation/reliability assessment instead to see, whether the action reported is true in terms of actions in place. If you keep action as exposure and try to see the temporal trend, it does not fit the standard epidemiological definition.

I really like the indicators selection, can you please make a table and show within each drivers what were the indicators (denominator, numerator) and source of data.

Result- The result are well presented especially with the graphs.

Discussion

Can you please method a section on methodological consideration, which is missing.

Reviewer #2: I must commend the ingenuity in applying regression statistical analysis to the DPSEEA framework to assess the association between national policies and trends in antibiotic resistance for 23 years across 73 countries. This approach not only adds a novel dimension to antibiotic resistance research but also provides valuable insights that could guide future interventions.

While your manuscript provides a comprehensive overview of the indicators and data sources, I noticed that specific datasets or versions are not always mentioned (e.g., Lines 125 and 160). Perhaps, you could state, 'The data used for antibiotic consumption trends were sourced from the QuintilesIMS MIDAS database, version X.Y (DOI: xxxx).'

Explicitly providing DOIs, URLs, and any conditions for data access would greatly enhance the transparency and reproducibility of your study, in line with PLOS’s data policy. Including these details will not only strengthen your manuscript but also make it easier for other researchers to build on your work.

Moreover, I recommend explicitly referencing the datasets where not already done (line 125 and 160) or including all of them in supplementary materials. This additional detail will further solidify your manuscript’s contribution to the field and ensure that it is a model of transparency in antibiotic resistance research.

6. PLOS authors have the option to publish the peer review history of their article (what does this mean? ). If published, this will include your full peer review and any attached files.

**Do you want your identity to be public for this peer review?** For information about this choice, including consent withdrawal, please see our Privacy Policy .

Reviewer #1: No

Reviewer #2: **Yes: ** Emmanuel Ifechukwude Benyeogor

---

## [Decision Letter · Decision Letter 1]

19 Mar 2025

Association between national action and trends in antibiotic resistance: an analysis of 73 countries from 2000 to 2023

PGPH-D-24-01686R1

Dear Dr. Søgaard Jørgensen,

We are pleased to inform you that your manuscript 'Association between national action and trends in antibiotic resistance: an analysis of 73 countries from 2000 to 2023' has been provisionally accepted for publication in PLOS Global Public Health.

Best regards,

Ashish KC, PhD

Academic Editor

Reviewer Comments (if any, and for reference):

Reviewer's Responses to Questions

**Comments to the Author**

1. If the authors have adequately addressed your comments raised in a previous round of review and you feel that this manuscript is now acceptable for publication, you may indicate that here to bypass the “Comments to the Author” section, enter your conflict of interest statement in the “Confidential to Editor” section, and submit your "Accept" recommendation.

Reviewer #1: All comments have been addressed

Reviewer #2: All comments have been addressed

2. Does this manuscript meet PLOS Global Public Health’s publication criteria ? Is the manuscript technically sound, and do the data support the conclusions? The manuscript must describe methodologically and ethically rigorous research with conclusions that are appropriately drawn based on the data presented.

Reviewer #1: Yes

Reviewer #2: (No Response)

3. Has the statistical analysis been performed appropriately and rigorously?

Reviewer #1: Yes

Reviewer #2: Yes

4. Have the authors made all data underlying the findings in their manuscript fully available (please refer to the Data Availability Statement at the start of the manuscript PDF file)?

Reviewer #1: Yes

Reviewer #2: Yes

5. Is the manuscript presented in an intelligible fashion and written in standard English?

Reviewer #1: Yes

Reviewer #2: Yes

6. Review Comments to the Author

Reviewer #1: The title and the abstract reads very well.

The introduction section in the main text reads very well.

The method section now provides research design, the settings, the data collection and extraction process.

The results have now been provided succinctly and intelligently.

The methodological consideration is now added and provides the limitation

Reviewer #2: Thank you for addressing my comments and your efforts in enhancing the transparency and clarity of the manuscript. The inclusion of explicit dataset references, citations, reordering of supplementary tables and improved descriptions further improve the manuscript's organization. These revisions will undoubtedly enhance the study’s impact and usability for future research in antibiotic resistance.

7. PLOS authors have the option to publish the peer review history of their article (what does this mean? ). If published, this will include your full peer review and any attached files.

**Do you want your identity to be public for this peer review?** For information about this choice, including consent withdrawal, please see our Privacy Policy .

Reviewer #1: No

Reviewer #2: No
